# The Effectiveness of the Integrated Care Model among Community-Dwelling Older Adults with Depression: A Quasi-Randomized Controlled Trial

**DOI:** 10.3390/ijerph19063306

**Published:** 2022-03-11

**Authors:** Su-Jung Liao, Shu-Mei Chao, Yu-Wen Fang, Jiin-Ru Rong, Chia-Jou Hsieh

**Affiliations:** 1Department of Nursing, College of Nursing, Tzu Chi University of Science and Technology, Hualien 970302, Taiwan; sc134@ems.tcust.edu.tw (S.-J.L.); yvonne@ems.tcust.edu.tw (Y.-W.F.); 2School of Nursing, National Taipei University of Nursing and Health Science, Taipei 112303, Taiwan; jiinru@gmail.com; 3Department of Nursing, Yuli Hospital, Ministry of Health and Welfare, Hualien 981, Taiwan; yl1292@ttyl.mohw.gov.tw

**Keywords:** depression, old patients, integrated care model, community, quasi-randomized controlled trial

## Abstract

Depression is the second-leading cause of disability among older patients worldwide. This study examined the effectiveness of the Integrated Care Model (ICM) intervention in decreasing depression, suicide ideation, and psychological symptoms and improving life satisfaction among old patients with depression living in communities. The participants were allocated to either the intervention (*n* = 67) or control group (*n* = 76) from July 2018 to November 2018. All participants received the usual geriatric care for three months in eastern Taiwan. Additionally, participants in the intervention group were treated according to the ICM, including the assessment and management of health problems, improvements in spiritual and mental wellbeing, helping with activities of daily life and mobility, providing social welfare resources, and prevention of elder abuse. The patients in the experimental group showed significant improvement in the group-by-time effect on the Center for Epidemiologic Studies Depression Scale, the Brief Symptom Rating Scale, suicide ideation measures, and the Life Satisfaction Index over 18 weeks. The ICM should be included in routine geriatric care and adopted in hospitals, long-term care facilities, and communities

## 1. Introduction

Depression is a serious global health threat as the number of cases continue to rise. Globally, depression affects an estimated 280 million people, with 5.7% being adults over the age of 60 [1]. According to a long-term follow-up survey of physical, mental, and social living conditions of middle-aged adults in Taiwan, the prevalence rate of depression in the elderly is about 15.1% [2]. Depression and depressive symptoms among older people may result in higher and higher medical and social costs. Since Taiwan has a higher prevalence rate than the rest of the world, providing appropriate care is a vital issue [3,4].

Depression is not a part of the natural aging process, although most older adults face this psychological problem. In addition, the incidence of depression tends to increase with age [5,6,7]. Depression is related to being bedridden, having unexplained physical symptoms, pain, and incapacity [6,8]. Studies revealed that depressive symptoms in older adults are related with disability [9,10]. The common causes of depression in the elderly include psychological conditions, social factors, and stress, and these increase with age. The physical function of the elderly will gradually decline, leading to an increase in the incidence of various acute and chronic diseases. Studies have found that the elderly in the community suffer from hypertension, chronic diseases such as diabetes, poorer perceived health, and higher chances of depression [11,12,13,14]. Retirement, changing social roles, a reduced financial capacity, and lost experiences also increase depression [15,16,17,18]. The other factors related to depression among the elderly in Taiwan were being female, having a low education level, living alone, being spouseless, financial difficulties, being jobless, and worsening physical health [7,12]. The higher the level of depression, the higher the amount of suicidal ideation, and the higher the rate of suicide among the elderly [11,19,20,21]. Due to aging, the elderly bear psychological and social pressures, physical diseases, and the inability to take care of themselves, which means that such a life change often leads to more depression. Life satisfaction among the elderly is one of the common predictors used to measure health in geriatric care [22]. Higher levels of education, better physical and psychological health status, more social support, better self-care, and better economic status were predictors of a higher life satisfaction trajectory [22,23]. Older adults with fewer stressful events, more stable emotions, better functioning in activities of daily living, better perceived health, lower depression, and fewer health problems have higher life satisfaction [11,24,25].

Since the growing number of older adults correlates with an increase in the number of depression cases, it was important to recognize depressive symptoms in the early stage, and perform appropriate intervention and treatments as soon as possible [5,26,27] for achieving better health. The interventions for decreasing depression include social support [28], exercise [29] and physical activities, reminiscence therapy, aid in daily activities, and adequate social resources [30,31]. Social support may protect older adults against worsening depression, and mitigate psychosocial stressors through emotional and tangible support. Healthcare professionals educate the elderly to help them adopt positive behaviors and strategies to manage depression so that patients can cope with stress, reduce negative thinking and depression, maintain mental health, and promote life adaptation [28,32,33]. Families provide emotional and tangible support to patients in performing daily activities, encouraging them to adhere to medical regimen, and accompany them to participate in social activities. Schuch and Stubbs (2019) indicated that exercise and physical activities increased the release of endorphins and reduced depression and depressive symptoms [30]. Ideal geriatric care includes a comprehensive integrated care system that gives the elderly the skills required for functional activities of daily life or assisted daily living and social needs [30]. Integrative reminiscence reduces depression and promotes life satisfaction and emotional wellbeing [30,31]. Older adults can adopt integrative reminiscence, including remembering and interpreting life events for recalling a memory, and rebuild the value of life and self-worth to achieve psychological wellbeing. Healthcare providers have sought social resources to help reduce the financial burden or provide the assistive devices needed for daily activities. In addition, healthcare professionals provide geriatric care, which includes treatments in hospitals, and medical home services. When patients encounter abuse or difficulties in life situations, healthcare professionals provide information about helplines or seek emergency shelter resources.

To sum up the above literature, the Integrated Care Model (ICM) combines case management, community care, and home care to provide intervention to older patients with depression and their families. In addition, a psychological professor, geriatric physicians and nurses, psychiatric doctors, social workers, occupational therapists, and caregivers have been involved in discussing the contents of the ICM. To sum up, the ICM includes assessing and managing health problems, achieving spiritual and mental wellbeing, enhancing activities of daily life and mobility, providing social welfare resources, the prevention of elder abuse, detecting depression at an early stage and delivering appropriate interventions for decreasing depressive conditions [11] and suicide ideation, and improving health status and life satisfaction. The ICM emphasizes patient-centered care and involves collaboration with families and professional healthcare providers to achieve better health [28,33].

The aim of this study was to examine the effects of intervention with the ICM on depression patients over time compared to the usual geriatric care. In addition, the study explored whether the intervention can decrease psychological symptoms and suicide ideation, as well as improve life satisfaction. The ideal intervention would be able to maintain its usefulness in the long term. Therefore, this study also examined if the long-term benefits of the intervention were sustained among the participants 6 weeks after the end of this intervention.

## 2. Materials and Methods

### 2.1. Participants and Eligibility Criteria

Study participants were selected from community-dwelling older adults using the following inclusion criteria:Above 55 years of age;Has a medical diagnosis of depressive disorder or depressive mood;Brief Symptom Rating Scale (BSRS-5) > 5;Mini-Mental State Examination (MMSE) > 20;Clear consciousness and attention sustainability of, at least, 20 min; depressed mood or symptoms do not interfere with interviews and data collection;Volunteering to participate in this study.

Since people with depression above 55 years of age meet the definition of older patients according to the health policy in Taiwan [34], participants over 55 years were recruited instead of 65 years. Sample exclusion criteria were:Visual, hearing, and communication impairments;Alcohol or drug dependence.

Eligible cases were referred by the attending physicians. A further diagnostic interview by a psychiatric doctor confirmed that 147 participants met the study criteria, and they were explained the study by the researchers. Two patients moved to live with their sons in other cities, and two others were hospitalized due to medical and surgical diseases. In total, 143 elderly patients voluntarily submitted informed written consent, and were recruited. Participants were rewarded with USD 3 gift coupons for participating in this study.

### 2.2. Study Design and Procedure

A quasi-randomized experiment, with repeated measures over 12 and 18 weeks, was conducted to determine the effect of the ICM on decreasing depression status (CES-D), improving psychological symptoms (BSRS-5), decreasing suicide ideation (SI), and improving the Life Satisfaction Index (LSI). All participants possibly share information and resources about the ICM interventions when they live in the same community, resulting in sample contamination effects. To decrease this possible influence, coin flipping was used to randomly assign two communities to determine which one was the experimental group. The intervention providers, the research assistants, and the patients were blind to this process. Consequently, 67 participants in the intervention group received the ICM intervention, while the 76 participants in the control group received the usual geriatric care by nurses during routine medical appointments at outpatients’ clinics and homes, such as checking lab data, measuring blood pressure, and adhering to taking medications. The nurses who specialized in psychiatric care have been working for at least three years providing care to participants in both groups. Additionally, nurses in the intervention group received training regarding the ICM intervention and then provided ICM interventions to participants, while nurses in the control group delivered the usual geriatric care. Two researchers collected data and were not responsible for providing care. The data were collected from July 2018 to November 2018.

This study was conducted according to the provisions of the Declaration of Helsinki (1995), and was approved by Yuli Hospital, Ministry of Health and Welfare Ethics Committee (YLH-IRB-10504). The process flowchart is illustrated in Figure 1.

### 2.3. Measures

#### 2.3.1. The Center for Epidemiologic Studies Depression Scale (CES-D)

The CES-D, originally published by Radloff in 1977, was designed for use in epidemiology studies to assess the degrees of depressive symptoms, and detect at-risk individuals among older adults. Response options ranged from 0 to 3 for each item (0 = Rarely or None of the time, 1 = Some or Little of the time, 2 = Moderately or Much of the time, 3 = Most or Almost all the time). Scores ranged from 0 to 60. The total scores had a 20-item measure that asked participants to rate how often, during the past week, they experienced symptoms associated with depression, such as restless sleep, poor appetite, and feeling lonely.

High scores indicated greater depressive symptoms. The CES-D has demonstrated reliability and validity across various Chinese populations, such as those who attempt suicide [35,36,37]. The study’s internal consistency Cronbach’s α was 0.82.

#### 2.3.2. Brief Symptom Rating Scale (BSRS-5)

This scale was first introduced by Lung [38], and developed by Professor Mingbin Li from the Department of Psychiatry, College of Medicine, National Taiwan University School (2003) [39]. The BSRS-5 measures the five symptom items of anxiety, depression, hostility, interpersonal sensitivity/inferiority, and insomnia and contains five items selected from the BSRS-50 that were highly correlated with each other, and included an additional question about suicide ideation (SI). Specifically, the BSRS-5 measured the following symptoms: difficultly in falling asleep, experiencing annoyance or anger, feeling down or depressed, a sense of inferiority to others, and having suicidal thoughts. The total score of the BSRS-5 is 24 points, with 0–4 points for each item (0: Not present, 1: Slight, 2: Moderate, 3: Severe, 4: Extremely severe). The internal consistency Cronbach’s α was 0.84 in this study.

#### 2.3.3. Life Satisfaction Index (LSI)

The LSI covers the general feelings of wellbeing among older people to identify “successful” aging. The concept of life satisfaction is closely related to morale, adjustment, and psychological wellbeing. Neugarten et al. (1961) [40] identified five components of life satisfaction that the LSI intended to measure. These included zest (as opposed to apathy), fortitude and resolution, congruence between achieved and desired goals, mood tone, and positive self-concept. The LSI was scored on a three-point scoring system for each item. Participants rated a satisfied response as 2, an uncertain response as 1, and an unsatisfied response as 0. Higher scores indicate better life satisfaction [40]. The study’s internal consistency Cronbach’s α was 0.75.

### 2.4. Interventions

Intervention providers graduated from a nursing college with a degree, and had more than three years of work experience. They are now serving in hospitals, outpatient clinics, communities, and other mental health-related institutions. The nurse providers received the ICM training for three months.

The ICM intervention is illustrated in Table 1. The intervention providers delivered the ICM to the participants in the intervention group. The participants in the control group received the usual geriatric care. All participants completed the questionnaires in the first week (before the intervention), at 12 weeks (at the end of the intervention), and at 18 weeks (6 weeks after the end of the intervention).

### 2.5. Analysis

Data were analyzed using descriptive statistics, the generalized estimating equation (GEE), and independent t-tests. The baselines of the demographic variables—CES-D, BSRS-5, SI, and LSI—of the two groups were compared using a t-test for continuous variables or χ^2^ for categorical variables. The GEE analyzed longitudinal data for outcomes and group-by-time interactions within, and between, group effects [41]. The GEE analysis used multiple repeated measures for continuous outcomes. The control group and baseline assessment were the reference points for performing GEE. The liner model was run with an auto-regressive first order, and post hoc analysis (Bonferroni test) was used to determine the trends of change in CES-D, BSRS-5, SI, and LSI at the baseline.

## 3. Results

### 3.1. Characteristics of Participants

A total of 143 participants were enrolled in the study, of which 67 patients in the intervention group and 76 patients in the control group completed the study for 18 weeks. The participant retention rate was 100% in both the groups (Figure 1).

The demographic characteristics of the participants in both groups at the baseline are presented in Table 2. In both groups, the majority of the participants were male (94, 65.7%), with the mean age of 68.85 years (*SD* = 1.38), more than half reported having an elementary education (101, 70.6%), the majority had never been married (100, 69.9%), and nearly half self-rated a poor economic status (71, 49.7%). The gender (χ^2^ = 6.18, *p* = 0.013), the economic status (χ^2^ = 9.15, *p* = 0.010), the SI (*t* = 2.33, *p* = 0.021), and the LSI (*t* = −4.13, *p* < 0.001) showed statistically significant differences between the two groups. The baseline values were adjusted for covariance in the GEE model. To understand the effectiveness of the ICM intervention, the GEE was computed only for those outcomes on which time and group had interaction effects, using gender, economic status, SI, and LSI where they had statistical significances for measures of covariance.

### 3.2. Results on Outcome Measures

Overall, CES-D, BSRS-5, SI, and LSI scores steadily improved in the intervention group, after controlling for covariance (Figure 2). The GEE revealed an interaction effect (Table 3) of Group × Time at 18 weeks (β = −4.76, 95% CI = −7.39~−2.14, *p* < 0.001) in CES-D; at 12 weeks (β = −1.79, 95% CI = −3.01~−0.57, *p* = 0.004) and at 18 weeks (β = −4.39, 95% CI = −5.60~−3.18, *p* < 0.001) in BSRS-5; at 12 weeks (β = 0.75, 95% CI = 0.45~1.05, *p* < 0.004) in SI; at 12 weeks (β = 2.72, 95% CI = 0.48 ~4.97, *p* = 0.018) and at 18 weeks (β = 5.06, 95% CI = 2.57~7.54, *p* < 0.001) in LSI.

More patients in the intervention group improved in between the time differences in CES-D at 12 weeks (β = −3.05, 95% CI = −4.68~−1.42, *p* < 0.001) and in BSRS-5 at 12 weeks (β = −1.79, 95% CI = −2.57~−1.01, *p* < 0.001).

## 4. Discussion

The results of this study indicate that the ICM can effectively decrease depression, psychological symptoms, and suicide ideation, as well as improve life satisfaction among older patients with depression living in communities. Additionally, the ICM has a long-term influence, even after the end of the intervention, on depression, psychological symptoms, and life satisfaction. In contrast, participants experienced worsening of conditions in the control group.

The results of this study were consistent with previous studies that indicated that social support from family, friends, or healthcare professionals was effective in decreasing depression, suicide ideation, and psychological symptoms, as well as improving life satisfaction [11,32,33]. Each domain of the ICM contained emotion or tangible social support. In Taiwan, the ICM was developed on the basis of a long-term care plan that adopted the integrated care system to care for elders with physical or mental disorders for achieving health and decreasing the burden on the families. Family and friends aided daily activities and encouraged patients to engage in activities in day care centers or community centers. Most older patients with depression in this study were never married. They either lived alone or with their family. In such scenarios, neighbors, neighborhood representatives, and volunteers also play a vital role in delivering tangible support for daily necessities or assistive devices. Healthcare professionals also encouraged the adherence to medical regimens and skills for reducing depression. Their supporters included family, friends, neighbors, volunteers, and neighborhood representatives to achieve the goals of their long-term care plans and independent living; “aging in a place” rather than “hospitalization”. Therefore, the social support of the ICM seemed more effective compared to the usual geriatric care.

Patients were also encouraged to exercise for at least 30 min, three to five days a week, which would release more endorphins, reduce depression, and promote physical vigor [29]. Moreover, people who exercised had more social interactions and reduced loneliness. Day care centers and community centers designed resistance and stretching exercises, and invited patients to engage. Exercise also slowed down the aging progress. Therefore, it is an important strategy for active aging.

Reminiscence therapy reduced depression symptoms and loneliness and also improved wellbeing and life satisfaction in older adults [31]. Healthcare professionals encouraged patients to share old photos, or listen to old songs to stimulate their memories, recall past events, feelings, and thoughts to achieve spiritual and mental wellbeing, inspire pleasure, reshape the impact of past experiences, and rebuild the value of life and self-worth.

Improving functional activities of daily living also reduced depression [42] and improved life satisfaction. The providers assessed functional activities of daily life and delivered daily activity training to ensure that patients could perform regular activities. For disabled elders, the Social Affairs Bureau or charities offered tangible support, such as assistive devices [43]. To assist independent living, neighborhood representatives and volunteers gave meals to those who could not prepare meals. On the basis of the long-term care policy in Taiwan, nurses and home care assistants provided regular aid by visiting homes and assisting in daily life. 

Elder abuse was related to depression. WHO (2021) [44] indicated, “Elder abuse includes physical, sexual, psychological, and emotional abuse; financial and material abuse; abandonment; neglect; and serious loss of dignity and respect.” Older adults who experience abuse tend to be more depressed. Elder abuse within families is a taboo topic in Asian countries. The excess work load of the caregivers sometimes results in neglect or rude behavior. The ICM provided referral respite care or day care centers for the elderly with depression and their caregivers. When elders felt threatened, the ICM also offered helpline numbers and emergency shelters for them.

Significant differences were noted in the status of mental health, suicidal ideation, and life satisfaction after a 12-week intervention of the ICM, and in depressive condition, mental health status, and life satisfaction, which was tracked for 6 weeks after the end of the intervention (at 18 weeks). It is evident that the intervention of the nursing staff can alleviate the mental health problems of the elderly and have a positive impact on improving depression in the elderly [45].

Providing the ICM for the older adults with depression can promote their health [11,24,46]. Therefore, the integrated care of this study was related to the development and promotion of a prevention and treatment system for depression, the early screening of the elderly with depression in the community, provision for appropriate care, and preventive measures to reduce the incidence of suicide so that the elderly can “age healthy and prevent disability,” which was consistent with the long-term care policy [43]. Findings show that integrated care interventions are, statistically and clinically, effective on health status, suicidal ideations, depressive status, and life satisfaction in community-based elderly patients with depression [9,10,47,48]. There are some limitations to this study. First, participants were recruited from a rural area in Taiwan, which may limit the generalizability of the research findings to different regions and cultural backgrounds. Secondly, using coin flipping to randomly assign two communities resulted in unequal group sizes. Some of the demographic and clinical characteristics at baseline were statistically different, such as gender and economic status, which led to potential sources of bias. Further studies should use an RCT design to examine the effectiveness of ICM. Therefore, more in-depth research is needed to explore the related factors affecting depression in the elderly, and more empirical intervention strategies are needed to reduce the occurrence of depression in the elderly.

## 5. Conclusions

After the ICM intervention for elderly patients with depression, there were significant differences between the two groups in terms of health status, suicidal ideations, depressive state, and life satisfaction. The ICM can maintain and improve the overall functional performance of the elderly. In future, in addition to requiring a longer research intervention, the integrated care model can be further extended to the elderly who are hospitalized or in long-term care institutions. It is hoped that the ICM will become universal, and will provide clinical and community mental health care professionals with a reference for care to improve the depressive symptoms of the elderly so that every elderly person be able to live and enjoy a healthy life, both in body and mind.

## Figures and Tables

**Figure 1 ijerph-19-03306-f001:**
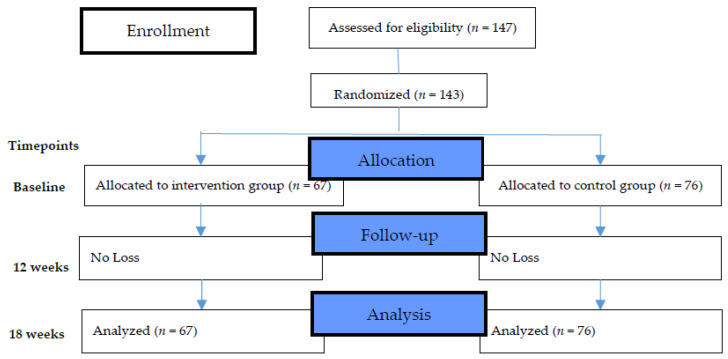
The flowchart of the research design and allocation of participants.

**Figure 2 ijerph-19-03306-f002:**
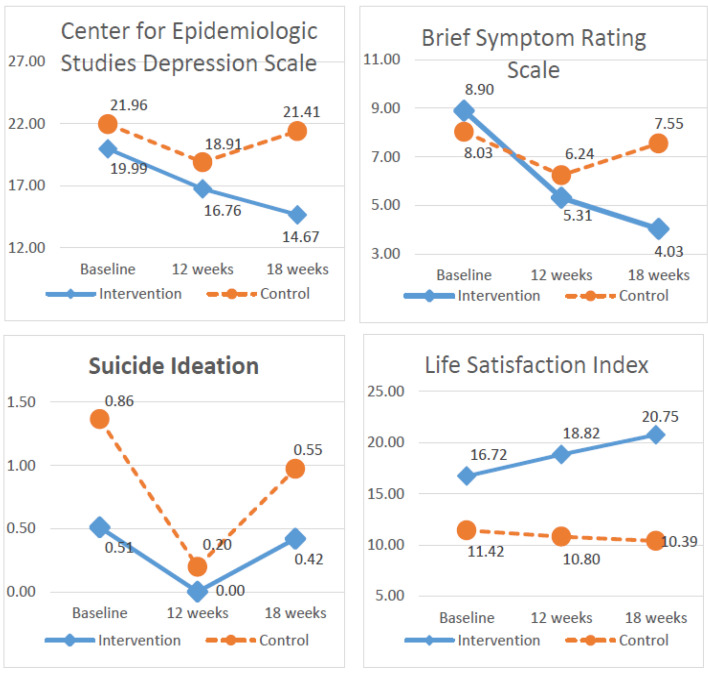
The trends for the Center for Epidemiologic Studies Depression, Brief Symptom Rating Scale, suicide ideation, and Life Satisfaction Index.

**Table 1 ijerph-19-03306-t001:** Interventions of Integrated Care Model.

Domains	Interventions
Assessing and managing health problems	To assess physical condition
	To manage physical health problems, such as chronic diseases
	To assess mental condition
	To manage mental health problems, such as dementia and delirium
	To assess dangerous behaviors, such as violence, self-injury, and psychiatric symptoms
	To provide problem-solving skills for dangerous behaviors, such as violence and self-injury. If necessary, consult psychologists or visit doctors
	To provide information about medication
	To encourage adherence to medication routines
	To collaborate with medical teams for patients’ care
Achieving spiritual and mental wellbeing	To offer emotional support
	To offer skills for stress management
	To encourage exercise
	To provide reminiscence therapy in group exercise
Enhancing activities of daily life and mobility	To assess functional activities of daily life
	To improve functional living skills, such as daily activity training
	To provide tangible support, such as assistive devices
	To arrange group exercise or individualized activities for facilitating physical activities
	To assist in living life independently, such as meal provision by neighborhood representatives or volunteers for those who cannot prepare meals
	To provide a safe environment to avoid falls and injuries
Providing social welfare resources	To inform social welfare systems
	To arrange and provide a rehabilitation bus for medical appointments
	To provide services at day care centers
	To provide respite care and short break services to decrease family burden
	To seek charity for financial difficulties
Providing prevention for elder abuse	To identity characteristics of elder abuse
	To relieve the burden on the caregiver, such as referral respite care or day care centers
	To provide helplines and emergency shelters, if necessary

**Table 2 ijerph-19-03306-t002:** Demographic and clinical characteristics at baseline.

Variable	Intervention Group (*n* = 67) *n* (%)	Control Group (*n* = 76) *n* (%)	Total (Percentage)	χ^2^/*t* *	*p*
Gender				6.18	0.013
Male	37 (55.2%)	57 (75.0%)	94 (65.7%)		
Female	30 (44.8%)	19 (25.0%)	49 (34.3%)		
Age				11.95	0.102
50–54 y/o	1 (1.5%)	0 (0.0%)	1 (0.7%)		
55–59 y/o	17 (25.4%)	11 (14.5%)	28 (19.6%)		
60–64 y/o	24 (35.8%)	20 (26.3%)	44 (30.8%)		
65–69 y/o	15 (22.4%)	21 (27.6%)	36 (25.2%)		
70–74 y/o	7 (10.4%)	12 (15.8%)	19 (13.3%)		
75–79 y/o	3 (4.5%)	5 (6.6%)	8 (5.6%)		
80–84 y/o	0 (0.0%)	6 (7.9%)	6 (4.2%)		
85–89 y/o	0 (0.0%)	1 (1.3%)	1 (0.7%)		
Education				7.30	0.121
Illiterate	0 (0.0%)	4 (5.3%)	4 (2.8%)		
Elementary school	45 (67.2%)	56 (73.7%)	101 (70.6%)		
Junior high school	9 (13.4%)	10 (13.2%)	19 (13.3%)		
Senior high school and above	11 (16.4%)	5 (6.6%)	16 (11.2%)		
Above college	2 (3.0%)	1 (1.3%)	3 (2.1%)		
Marital status				7.92	0.095
Never married	47 (70.1%)	53 (69.7%)	100 (69.9%)		
Widow/widower	6 (9.0%)	5 (6.6%)	11 (7.7%)		
Live-in together	2 (3.0%)	0 (0.0%)	2 (1.4%)		
Married	3 (4.5%)	0 (0.0%)	3 (2.1%)		
Separated/divorced	9 (13.4%)	18 (23.7%)	27 (18.9%)		
Economic status				9.15	0.010
Poor	26 (38.8%)	45 (59.2%)	71 (49.7%)		
Fair	37 (55.2%)	23 (30.3%)	60 (42.0%)		
Sufficient	4 (6.0%)	8 (10.5%)	12 (8.4%)		
Health status				−1.54 *	0.127
Very bad	2 (3.0%)	6 (7.9%)	8 (5.6%)		
Bad	15 (22.4%)	21 (27.6%)	36 (25.2%)		
Good	37 (55.2%)	38 (50.0%)	75 (52.4%)		
Very good	13 (19.4%)	11 (14.5%)	24 (16.8%)		
Face scale mood				0.96 *	0.337
Crying	7 (10.4%)	0 (0.0%)	7 (4.9%)		
Very sad	3 (4.5%)	2 (2.6%)	5 (3.5%)		
Somewhat sad	11 (16.4%)	25 (32.9%)	36 (25.2%)		
Neutral	28 (41.8%)	33 (43.4%)	61 (42.7%)		
Somewhat happy	14 (20.9%)	8 (10.5%)	22 (15.4%)		
Very happy	3 (4.5%)	5 (6.6%)	8 (5.6%)		
Extremely happy	1 (1.5%)	3 (3.9%)	4 (2.8%)		
	Mean (*SD*)	Mean (*SD*)	Mean (*SD*)		
Health status	1.91 (0.73)	1.71 (0.81)	1.80 (0.78)	−1.54 *	0.127
Chronic illness number	1.81 (1.61)	1.43 (1.24)	1.61 (1.43)	−1.56 *	0.121
CES-D	19.99 (7.89)	21.96 (8.74)	21.03 (8.38)	1.41 *	0.160
BSRS-5	8.90 (3.05)	8.03 (2.14)	8.43 (2.63)	−1.95 *	0.054
SI	0.51 (0.89)	0.86 (0.89)	0.69 (0.91)	2.33 *	0.021
LSI	16.72 (7.81)	11.42 (7.49)	13.9 (8.06)	−4.13 *	<0.001

Note: CES-D: Center for Epidemiologic Studies Depression Scale; BSRS-5: Brief Symptom Rating Scale; SI: suicide ideation; LSI: Life Satisfaction Index; * = *t* value.

**Table 3 ijerph-19-03306-t003:** Results of generalized estimation equations in Center for Epidemiologic Studies, Brief Symptom Rating Scale, suicide ideation, and Life Satisfaction Index.

Parameters	Center for Epidemiologic Studies Depression Scale			Brief Symptom Rating Scale		
	β	95% CI	*p*	β	95% CI	*p*
Intercept	21.33	16.51~26.15	<0.001	7.90	6.55~9.25	<0.001
Gender	1.38	−1.04~3.80	0.265	−0.04	−0.99~0.92	0.941
Economic status	0.25	−1.73~2.23	0.804	−0.05	−0.74~0.64	0.890
Suicide ideation	1.40	0.07~2.74	0.039	0.82	0.15~1.50	0.016
Life Satisfaction Index	−0.11	−0.26~0.03	0.131	−0.04	−0.09~0.01	0.082
Group ^a^	−1.12	−3.67~1.43	0.389	1.38	0.56~2.21	0.957
Time						
12 weeks ^b^	−3.05	−4.68~−1.42	<0.001	−1.79	−2.57~−1.01	<0.001
18 weeks ^b^	−0.55	−2.38~1.27	0.553	−0.47	−1.26~0.31	0.236
Group * Time						
Intervention * 12 weeks ^c^	−0.17	−2.40~2.06	0.880	−1.79	−3.01~−0.57	0.004
Intervention * 18 weeks ^c^	−4.76	−7.39~−2.14	<0.001	−4.39	−5.60~−3.18	<0.001
**Parameters**	**Suicide Ideation**			**Life Satisfaction Index**		
	**β**	**95% CI**	** *p* **	**β**	**95% CI**	** *p* **
Intercept	1.36	0.91~1.81	<0.001	17.30	13.60~20.99	<0.001
Gender	−0.13	−0.36~0.10	0.265	−0.46	−2.21~1.29	0.605
Economic status	−0.09	−0. 28~0.09	0.334	−2.75	−4.50~−1.01	0.002
Suicide ideation	-	-	-	−1.95	−2.88~−1.02	<0.001
Life Satisfaction Index	−0.03	−0.04~0.02	<0.001	-	-	-
Group ^a^	−0.18	−0.49~0.13	0.248	4.27	1.87~6.67	<0.001
Time						
12 weeks ^b^	−0.66	−0.86~−0.45	<0.001	−0.62	−2.52~1.28	0.523
18 weeks ^b^	−0.30	−0.54~−0.07	0.012	−1.03	−3.09~1.04	0.331
Group * Time						
Intervention * 12 weeks ^c^	0.75	0.45~1.05	<0.001	2.72	0.48~4.97	0.018
Intervention * 18 weeks ^c^	0.21	−0.10~0.53	0.186	5.06	2.57~7.54	<0.001

Note: ^a^ Reference group: control group. ^b^ Reference group: baseline (time). ^c^ Reference group: control group * baseline.

## Data Availability

Not applicable.

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
