# Peer review of "The Effectiveness of the Integrated Care Model among Community-Dwelling Older Adults with Depression: A Quasi-Randomized Controlled Trial"

_ijerph, 2022, doi:10.3390/ijerph19063306_

Round 1

Reviewer 1 Report

Page 2 - citation Schuch & Stubbs (2019) must be corrected.

Results 3.1 both the groups syntagma must be replaged with both groups (eliminate THE).

  1. What is the main question addressed by the research?

Decreasing depression and suicidal ideation, as well as increasing life satisfaction as a result of participating in the ICM program

  1. Do you consider the topic original or relevant in the field, and if
    so, why?

It is not an original topic, but it is relevant because it reinforces the idea that older people need special attention and care to age well. Old age brings with it a number of shortcomings: impaired physical health, impaired body function, which affects a person's functionality in all areas of life. All this is corroborated with feelings of loneliness, uselessness, withdrawal from social life. Every person has the right to age beautifully and healthily.

  1. What does it add to the subject area compared with other published
    material?

The inclusion criteria for people between the ages of 55 and 65, which is specific to Taiwan.

  1. What specific improvements could the authors consider regarding the
    methodology?

The program should be described in detail, possibly a session attended by the experimental group. Also, a distinction must be made between the ICM program and the usual geriatric care.

  1. Are the conclusions consistent with the evidence and arguments
    presented and do they address the main question posed?

Yes

  1. Are the references appropriate?

Yes

  1. Please include any additional comments on the tables and figures.

Due to the large number of sociodemographic variables, the tables on descriptive statistics are too extensive, perhaps they should be condensed. The tables of results should be separated according to the outcome variables for a clearer representation of the results.

Strengths - the researchers maintained the same number of participants throughout the study (18 months).

Weaknesses – the authors may have had to present the results according to the age groups (for example 55-65, 66-75, 76-85 etc). It would have been interesting to see when exactly the ICM program can be introduced into standard geriatric care programs.

Author Response

Response to Reviewer

Thank your suggestions. We highlight the changes to the manuscript within the document by using red-colored text. Please see the revisions or responses in the manuscript and the following tables.

Reviewer: 1

Comments

Revisions or responses

I don't feel qualified to judge about the English language and style.

This manuscript was edited by Editage Academic Editing Company on January 29, 2022.

Page 2 - citation Schuch & Stubbs (2019) must be corrected.

Correct the reference number [30].

both the groups syntagma must be replaged with both groups

Deleted the.

What is the main question addressed by the research?

Decreasing depression and suicidal ideation, as well as increasing life satisfaction as a result of participating in the ICM program

Please see the 101-104 line. The aim of this study was to examine the effects of the ICM intervention on depression patients over time, compared to the usual geriatric care.

Yes, according to results, the ICM program can decrease psychological symptoms and suicide ideation, as well as improve life satisfaction.

Do you consider the topic original or relevant in the field, and if so, why?

It is not an original topic, but it is relevant because it reinforces the idea that older people need special attention and care to age well. Old age brings with it a number of shortcomings: impaired physical health, impaired body function, which affects a person's functionality in all areas of life. All this is corroborated with feelings of loneliness, uselessness, withdrawal from social life. Every person has the right to age beautifully and healthily.

Yes, we agree with the reviewer’s opinion, it is a relevant topic but it is an original research article.

We hope the ICM can be further extended to hospitalized, or long-term care institutions for achieving the best health for the elder.

What does it add to the subject area compared with other published material?

At present, the care services for patients with depression include outpatient medical care, inpatient medical care, daycare and home services, etc., but they cannot meet the care needs of the elderly with depression in the community and provide holistic care for the elderly with depression. This study firstly explores the care needs of the elderly with depression in the community, constructs a holistic care model for the elderly with depression in the community by means of action research, and then provides appropriate holistic care services for the elderly with depression in the community.

The inclusion criteria for people between the ages of 55 and 65, which is specific to Taiwan.

In this study, Hualien county is the investigated area, where the highest percentage of aboriginal people live in Taiwan. According to long-term care policy 2.0 by Executive Yuan in Taiwan, indigenous people over 55 years old are older adults who meet the research criteria. 

What specific improvements could the authors consider regarding the methodology?

Added the limitations the 384-391 line.

The program should be described in detail, possibly a session attended by the experimental group. Also, a distinction must be made between the ICM program and the usual geriatric care.

The contents of the ICM program were demonstrated in Table 1. Interventions of Integrated Care Model, while the usual geriatric care added in detail the 142-143 line.

Are the conclusions consistent with the evidence and arguments presented and do they address the main question posed?

Yes

Thank you so much.

Are the references appropriate?

Yes

Thank you so much.

Please include any additional comments on the tables and figures.

Due to the large number of sociodemographic variables, the tables on descriptive statistics are too extensive, perhaps they should be condensed.

The tables of results should be separated according to the outcome variables for a clearer representation of the results.

Thank your suggestions. Age should be numerical instead of categorical data, which look like redundant. It is difficult to retrospective the original age data as categorical data, therefore, it made the demographic data are not concise. In the future, those disadvantages will be modified in future studies.

Thank your precious suggestion. We make some modifications in Table 3 and hope those changes can be clearer.

Strengths - the researchers maintained the same number of participants throughout the study (18 months).

Thank you so much.

Weaknesses – the authors may have had to present the results according to the age groups (for example 55-65, 66-75, 76-85 etc). It would have been interesting to see when exactly the ICM program can be introduced into standard geriatric care programs.

Thank your suggestions. In the future, if we progress similar research, we will be explored and predict the age influences ICM. The aim of this study is to explore the effectiveness of ICM. Based on the assumptions of the generalized estimating equation, the baseline data should be considered as covariates when there were statistical differences. Because there were no statistical differences in age, we did not focus and explain.

Reviewer 2 Report

Attached the file with comments and corrections.

Author Response

Response to Reviewer

Thank your suggestions. We highlight the changes to the manuscript within the document by using red-colored text. Please see the revisions or responses in the manuscript and the following tables.

Reviewer: 2

comments

Revisions or responses

FORMAT

The title of the figures is placed below.

The title of the tables is placed above.

Revised.

Abstract

-Delete the statistical data of the abstract. In this section, the reader wants to know the

effectiveness of the intervention, so I consider that the description is sufficient, without numerical

data. The reader interested in the data analysis process will consult this information in the

corresponding section of the manuscript.

Deleted the numerical data.

2.1. Participants and Eligibility criteria

-If one of the objectives of the research is to reduce depression or depressive symptoms, I consider that it is essential to include the instrument that allows evaluating depression.

It's not included. Psychiatric diagnosis is indicated to identify depression.

-Add the name of the authors and the year in each instrument.

The research adopted Brief Symptom Rating Scale to measure the depressive status. We added the explanations of the instrument. In addition, we investigated depressive status by “2.3.1. The Center for Epidemiologic Studies Depression Scale (CES-D)” the 165-176 line.

Added.

2.2. Design and Procedure I consider that all instruments should be identified with the authors and the year in parentheses, adding the full name the first time they are mentioned: SI and LSI instruments. Similar to MiniMental State Examination (MMSE).

Revised.

Figure 1. In Baseline add the number of participants. Title below

Thank you for your kind reminding. Revised.

References

- Check the references. Some do not follow the format of the magazine. The name of the journal in

abbreviated form, for example, in the reference number 26: J Healthc Manag

Revised.

- In reference number 36. Follow the format of the previous references

Revised.